# Metabolizable Energy of Soybean Meal and Canola Meal as Influenced by the Reference Diet Used and Assay Method

**DOI:** 10.3390/ani10112132

**Published:** 2020-11-17

**Authors:** Shravani Veluri, Oluyinka Abiona Olukosi

**Affiliations:** Department of Poultry Science, University of Georgia, Athens, GA 30602, USA; shravani.veluri@uga.edu

**Keywords:** canola meal, soybean meal, reference diet type, metabolizable energy assay

## Abstract

**Simple Summary:**

A standard nutritionally balanced diet is used as a reference diet into which test feedstuffs are added to determine metabolizable energy of the added test feedstuff either by difference or regression method. We hypothesized that if the composition of the reference diet changes, the metabolizable energy of the test feedstuff will also changes and that the difference and regression methods will not give similar metabolizable energy values for the test feedstuff under these circumstances. To test this hypothesis, we used two reference diets (corn-soybean meal and corn-canola meal), instead of using just one reference diet, to determine metabolizable energy of soybean meal and canola meal. The metabolizable energy of soybean meal and canola meal was greater when corn-canola meal was the reference diet compared with using corn-soybean meal reference diet. Compared with the difference method, the regression method gave greater metabolizable energy value for the test feedstuffs. Results from the current study show that the calculated metabolizable energy value of feedstuffs depends on the reference diet used and that the difference and regression methods did not give similar metabolizable energy value for the test feedstuffs.

**Abstract:**

A 21-day experiment was conducted to study the effect of reference diet type and assay method on apparent metabolizable energy (AME) and nitrogen-corrected (AMEn) of soybean meal (SBM) and canola meal (CM). Broilers (*n* = 240) were allocated to 10 treatments with eight replicates/treatment and three birds/replicate. Treatments included corn-SBM or corn-CM reference diets (RD). To each RD, 300 or 450 g/kg of SBM or CM were added to make a total of eight test diets. For the difference method, AME of SBM and CM substituted at 300 g/kg in corn-CM RD gave greater AME values compared to inclusion in the corn-SBM RD. The AMEn of SBM increased with increase in inclusion level in the corn-CM RD but AMEn of CM decreased with increased inclusion level of CM in the corn-SBM RD. For the regression method, AME and AMEn of the test feedstuffs were greater with corn-CM RD compared with corn-SBM RD. The AME of SBM was not affected by assay method, whereas AME of CM was lower when determined using the regression method. In conclusion, both the reference diet type and assay method influenced assayed AME and AMEn value of test protein feedstuffs and should be considered in cross-studies comparisons.

## 1. Introduction

Feed represents 70% of the total cost of broiler production, of which the energy represents the greatest proportion. Available energy from feedstuffs in broilers is determined using apparent metabolizable energy (AME) and as nitrogen corrected apparent metabolizable energy (AMEn). Accurate determination of AME value of feedstuffs and using these values in feed formulations are considered some of the most important criteria required to improve feed efficiency and reduce feed costs [1,2]. However, there is wide variability in the reported AME and AMEn values for most feedstuffs making interpretation of the values and usage in feed formulation very challenging. Variation in reported AME values are often due to genetic factors, growing conditions, age, dietary factors and methods used in determining these AME values [3,4,5]. Over the years, several changes have been made in the experimental design and assumptions made during AME calculation [6,7,8,9,10]. In addition, there are various methods used to determine the AME and AMEn of feedstuffs, among which methods basal diet substitution and multiple linear regression method are regularly employed.

In basal diet substitution or multiple linear regression methods, the key is to formulate a reference diet which is a complete and nutritionally balanced diet. Test feedstuffs are included in the basal diets at different levels to make test diets so that AME can be determined either by difference or the regression method [11,12]. Recent studies showed that the AME of test feedstuffs depends on the composition of the reference diet used. As the composition of the reference diet changes, so does the AME of test feedstuff. For example, in a study reported by Olukosi et al., (2010) [10] adding graded levels of wheat to the reference diet reduced the AME of meat and bone meal by 5%.

Soybean meal (SBM) and canola meal (CM) are the two most commonly used plant protein feedstuffs for broilers. Because these protein feedstuffs constitute almost 30% of the broiler diets, it is essential to characterize the various factors that might affect their AME and AMEn values especially regarding the assay method or basal diets used in the assay. This is especially so because incorporation of additional SBM or CM, for example, dramatically changes the protein content of the test diets, and this may be dependent on type of feedstuffs used in the basal diet. The objective of the current study was to determine the effect of the reference diet type (CM or SBM based), assay method (difference or regression), and inclusion levels of the test feedstuff in the assay diet, on the AME and AMEn of SBM and CM.

## 2. Materials and Methods

### 2.1. Birds and Diets

The study was approved by the Institutional Animal Care and Use Committee at the University of Georgia (Athens, GA, USA; IACUC number: A2018 08-026). A total of 240 male broilers at zero-day old were used for the study. Birds received a standard pre-experimental diet based on 56% wheat, 20% SBM, and 15% CM from day 0 to 14. On day 14, birds were allocated to 10 treatments with eight replicates per treatment and three birds per replicate. The 10 treatments comprised corn-SBM and corn-CM reference diets (Table 1), and for each reference diet, four additional test diets were mixed consisting of two substitution levels (300 or 450 g/kg) of each of SBM or CM (Table 2). Titanium dioxide (TiO_2_) was added as an indigestible marker into each diet to determine nutrient utilization by the index method. The experimental diets were fed from days 14 to 21. Birds were given ad-libitum access to feed and water. Body weight and feed intake data were collected on days 14 and 21. Excreta were collected on days 20–21.

### 2.2. Chemical Analysis

Diets and excreta samples were ground through a 0.5-mm screen and analyzed for dry matter, gross energy, nitrogen, and TiO_2_ concentration. For dry matter determination, samples were dried at 105 °C in a drying oven for 24 h (AOAC Method 934.01; AOAC, 1990). Gross energy was determined using an isoperibol bomb calorimeter (Model 6200, Parr Instruments, Moline, IL, USA) using benzoic acid as a calibration standard. Total nitrogen content was determined by the combustion method (Method 968.06; AOAC, 1990). TiO_2_ concentration was measured using the method proposed by Short et al., (1996) [13]. Diets and excreta samples were ashed in furnace at 600 °C for 12 h. Ash was dissolved in 7.4 M sulfuric acid to release titanium. Hydrogen peroxide was added to oxidize titanium and absorbance was measured at 410 nm on a spectrophotometer.

### 2.3. Calculations

The metabolizable energy of diets was measured using the index method. Apparent metabolizable energy (AME, MJ/kg) was calculated subsequently by using the formula:$$AME=\;GE_{i}-\left[\;GE_{o}\times \;\left(\frac{C_{i}}{C_{o}}\right)\right];$$
where: *GE_i_* = Gross energy (MJ/kg) of the diet, *GE_o_* = Gross energy (MJ/kg) of the excreta, *C_i_* = Concentration of titanium (%) in the diet and *C_o_* = Concentration of titanium (%) in the excreta

Nitrogen-corrected apparent metabolizable energy (AMEn, MJ/kg) was calculated by using 34.4 kJ/g as a correction factor.

The AME and AMEn of test feedstuffs (SBM and CM), using either of the reference diet types (corn-SBM or corn-CM) were calculated using both the difference and the regression methods. For the difference method, the AME or AMEn of test feedstuffs were calculated after correcting the contribution of test feedstuff in the test diet for the fractional contribution of energy by the test feedstuffs into the test diets. The AME or AMEn of test feedstuffs were calculated using the formula:$$AME_{tf}=\;\frac{AME_{td}-\left(AME_{rd}\;\times \;P_{rd}\right)}{P_{tf}};$$
where: *AME_tf_* = AME or AMEn (MJ/kg) of test feedstuff, *AME_td_* = AME or AMEn (MJ/kg) of test diet, *AME_rd_* = AME or AMEn (MJ/kg) of the reference diet, *P_rd_* = Proportional contribution of the energy of the reference diet in the test diet and *P_tf_* = Proportional contribution of the energy of test feedstuff in the test diet.

For calculation of AME or AMEn of SBM and CM by the regression method, SBM- or CM-associated AME or AMEn intake in MJ were regressed against SBM or CM intake in kilograms. The AME or AMEn values of the test feedstuffs were the slope of the regression equations. Consequently, there were eight AME or AMEn values for each feedstuff corresponding to the number of replicates per treatment.

### 2.4. Statistical Analysis

All of the statistical analyses were conducted using JMP (JMP pro version 15.) The data for AME and AMEn of test feedstuffs calculated by the difference method were analyzed as a 2 × 2 × 2 factorial to establish the effect of reference diet type (corn-SBM or corn-CM), protein feedstuff (SBM or CM) and test feedstuff substitution levels (300 or 450 g/kg). The data for AME and AMEn of test feedstuffs calculated by the regression method were analyzed as a 2 × 2 factorial to show the effect of reference diet type (corn-SBM or corn-CM) and protein feedstuff (SBM and CM). The AME and AMEn data calculated using regression and difference methods were compared using a 2 × 2 × 2 factorial arrangement for delineation of the effect of reference diet (corn-SBM or corn-CM), protein feedstuff (SBM or CM), and assay method (Difference or Regression). The main effect means were discussed when there are no significant interactions. Simple effects are discussed when two- or three-way interactions, as appropriate, are significant. Significantly different means (*p* ≤ 0.05) were separated using Tukey’s HSD.

## 3. Results

### 3.1. AME and AMEn of Protein Feedstuffs Determined Using the Difference Method

There was no significant two- or three-way interaction for AME (Table 3). The AME of test feedstuffs, when determined using the corn-CM reference diet, was greater (*p* < 0.01) than when determined using the corn-SBM reference diet. In addition, AME of CM was greater than AME of SBM, and the AME was greater (*p* < 0.01) when test feedstuffs were substituted at 300 g/kg compared to substitution at 450 g/kg. There was a significant (*p* < 0.05) reference diet × feedstuff × substitution level interaction for AMEn calculated by the difference method. The AMEn of SBM was not influenced by an increasing inclusion level of SBM in the corn-SBM reference diet whereas AMEn of SBM increased (*p* < 0.05) as the inclusion level of SBM increased in corn-CM reference diet. On the other hand, the AMEn of CM decreased (*p* < 0.05) with increased inclusion level of CM in the corn-SBM reference diet, whereas there was no effect of CM inclusion level when AMEn was determined in the corn-CM reference diet.

### 3.2. AME and AMEn of Protein Feedstuffs Determined Using the Regression Method

There was no significant reference diet × feedstuff interaction for AME or AMEn determined by the regression method (Table 4). The AME and AMEn of test feedstuffs, when determined using the corn-CM reference diet, was greater (*p* < 0.01) than when determined using the corn-SBM reference diet, and AME of SBM was greater (*p* = 0.05) than CM. There was no significant effect of protein feedstuff on AMEn.

### 3.3. Effect of the Assay Method on AME and AMEn of Protein Feedstuffs

There was no significant three-way interaction between factors for AME and AMEn but there was significant protein feedstuff × method interaction (*p* < 0.01) for AME (Table 5). There was no significant difference between the AME of SBM calculated by the difference and the regression methods but the AME of CM calculated by the difference method was greater (*p* < 0.01) than the regression method. The AME and AMEn of protein feedstuffs, when determined using the corn-CM reference diet was greater (*p* < 0.01) than when determined using the corn-SBM reference diet. Overall the AME and AMEn of SBM were not significantly different.

## 4. Discussion

Metabolizable energy is the most commonly used expression for evaluating energy bioavailability from feed and feedstuffs for poultry. In determining the AME of feedstuffs, a nutritionally balanced diet is used as reference diet into which test feedstuffs are incorporated at different levels and AME is determined either by difference or regression method, among other indirect methodologies. Two important questions are these: What if the composition of the reference diet changes, does it still give similar AME and AMEn values for the test feedstuffs? Will the changes be influenced by the mathematical treatment of the assay data? The objective of the current experiment was to test the effect of composition of reference diet on determined AME and AMEn of SBM and CM, and hence answer the two questions. In order to achieve this, SBM or CM were incorporated at 300 or 450 g/kg into either of two reference diets (corn-SBM or corn-CM reference diets). This allowed AME and AMEn of both SBM and CM to be determined in both corn-SBM as well as corn-CM reference diets. The major differences between the two reference diets were in their primary protein feedstuff (SBM in corn-SBM reference diet and CM in corn-CM reference diet) and marginal difference in proportion of energy-yielding feedstuffs (953 and 946 g/kg in corn-SBM and corn-CM reference diets, respectively). The two reference diets were formulated both to be similar in energy and nutrient contents, including vitamins and trace minerals, and to meet nutritional requirement for Cobb 500 broiler chickens. In addition, the pre-experimental diets included both SBM and CM in order to ensure that the birds were exposed to consuming both protein feedstuffs from the start of the experiment.

### 4.1. Effect of the Assay Method and Reference Diet Type on AME and AMEn of the Protein Feedstuffs

The AME for SBM and CM was greater when included in corn-CM reference diet compared to inclusion in corn-SBM reference diet. The different values of metabolizable energy observed for the same feedstuff when included in two reference diets was likely due to interaction with test feedstuff and its metabolizable energy [10]. When test feedstuffs, SBM and CM, were added into the reference diets used in the current experiment, the crude protein (CP) content of test diets with corn-SBM as reference diet was greater than CP content of the diets with corn-CM as reference diets. Adding the test feedstuffs (CM or SBM), at the same inclusion levels, into the two reference diets resulted in lower dietary protein level in corn-CM reference diet compared with corn-SBM reference diet.

The dramatically higher dietary CP content observed in test diets with corn-SBM reference diets compared to corn-CM reference diets has implication on energy utilization. Nieto et al. [14] suggested that CP content of diets up to 23% can be well utilized by broilers without having negative effects on the metabolizable energy, but higher CP content led to excretion of the excess N with significant energy expenditure. Clearly, energy excretion will be higher for diets with higher CP content than the usual 20–23% CP, which will translate to lower energy availability and consequently lower AME. In agreement with this, nitrogen and energy excretion from diets with corn-SBM as reference diet were greater when compared to diets with corn-CM as reference diet in the current study. Therefore, this partly explained why the AME of test feedstuffs included in corn-SBM reference diet was less than AME of test feedstuffs with corn-CM as reference diet.

With particular reference to the difference method, the AME of SBM and CM decreased with increase in their inclusion levels in assay diets. This is expected because in test feedstuffs with high CP level, variation in inclusion level of the test feedstuff can dramatically alter the dietary protein level with concomitant influence on dietary AME [15,16,17]. Diets with SBM or CM substituted at 300 g/kg into the reference diet had lower CP content than the diets with substituted SBM or CM at 450 g/kg. This increase in dietary CP, as mentioned earlier, resulted in higher nitrogen and energy excretion ultimately leading to lower AME of the test feedstuff. A similar effect of reduction in AME with increase in inclusion level of SBM from 20 to 30% was reported by Lopez et al. [17] wherein AME of SBM 20% inclusion level in the reference diet gave greater AME compared with 30% inclusion. This decrease in AME with an increase in inclusion level for CM was also observed in some other studies when CM inclusion increased from 10 to 20% [18,19]. In the current study, it was possible to study the effect of increasing the test feedstuffs in two reference diet types simultaneously. The observation was that the decrease in AME with increase in inclusion level of high-protein feedstuffs was independent of reference diet used or test feedstuff being evaluated.

Another possible factor that may confer differences on AME of feedstuffs determined in CM or SBM reference diets is the anti-nutritional factors (ANF) in the feedstuffs and their interaction with test feedstuffs. For example, it is anticipated that high fiber and glucosinolates in CM, compared with SBM, may influence feed intake [20,21,22], and consequently reduce AME of test feedstuffs. In the current study, there was no effect of reference diet type on feed intake. Therefore, it appears that this factor was not influential in the current experiment. In addition, it has been shown that there was no correlation between AME and glucosinolate content of meals of modern breeds of canola [18].

The other possible reason for the observed greater metabolizable energy value when corn-CM was used as reference diet in the current experiment is likely due to ether extract content of the test feedstuff. The CM used for this study had gross energy of 19.7 MJ/kg, this amount of gross energy in CM, which was greater than usual, can be linked to higher ether extract content [23]. The ether extract in the test CM, and in feedstuffs generally, can improve AME and AMEn of a diet by reducing the passage rate of the digesta which can lead to increased energy availability [18,24,25]. This also partly explains the observed greater AME and AMEn values of SBM and CM when determined in the corn-CM reference diet compared to the corn-SBM reference diet.

The AMEn of SBM and CM was influenced by both the reference diet used and level of inclusion of test feedstuffs into the reference diets. Applying nitrogen correction to feedstuffs AME gives an estimate of comparable available energy from feedstuffs with different CP content [26]. So, it was expected that even if the test diets differ in their CP content and give different AME values for the same feedstuffs at different inclusion levels, correcting for N will make adjustments for the energy retained as protein. The observation from the current study indicate that this may not be the case when reference diet has the same protein feedstuff and test protein feedstuff (i.e., SBM being added to e.g., corn-SBM, rather than corn-CM reference diet).

For the AMEn of SBM, the inclusion level of the test feedstuff had no effect on assayed AMEn when included in corn-SBM reference diet. On the other hand, when included in corn-CM reference, assayed AMEn of SBM increased with increase in inclusion level. In contrast, assayed AMEn of CM was not influenced by inclusion level in corn-CM reference diet, whereas AMEn of CM decreased with an increase in inclusion level of CM in corn-SBM reference diet. The effect of inclusion level on nitrogen retention has been reportedly due to the combined effect of quality of test protein feedstuff, age of birds, and inclusion levels used [17,26]. Unlike AME, AMEn of test feedstuffs specifically for protein feedstuffs is more sensitive to quality of the reference diet used and level of inclusion of the test protein feedstuff into these reference diets.

### 4.2. Comparison of the Effect of Assay Method of AME and AMEn of the Protein Feedstuffs

The mathematical treatment of AME and AMEn data were also investigated in the current study. Difference and regression methods are two commonly used methods for calculation of metabolizable energy of feedstuffs but only few studies actually did a comparison of AME and AMEn of feedstuffs when determined by both methods [24]. Metabolizable energy as determined by difference method assumes that the test feedstuff and the feedstuffs in reference diet do not interact with each other and will be additive. Therefore, AME and AMEn values for test feedstuffs should give similar estimates irrespective of the composition of the reference diet used. It is recognized that there are still many undefined issues in AME assays [3,5]. The focus in the current study was on comparing AME and AMEn of SBM and CM when determined both by difference and regression methods and using different reference diets. The AME of CM was greater when determined using the difference method compared to the regression method. In contrast, AMEn of SBM and CM was greater when determined using the regression method compared to difference method. That observation may be related to the effect of nitrogen correction in the test diets with excess nitrogen content. The take-away from the results of the current study was that regression method gave a more consistent value for AME and AMEn compared with the difference method. This can be seen from the comparatively lower standard error values for AME and AMEn determined by the regression method compared with the difference method similar to observation of Tillman and Waldroup [27].

## 5. Conclusions

The conclusion from the observations in the current experiment was that AME of SBM depended on the reference diet and inclusion level used in the assay, whereas AMEn was influenced primarily by the method used in calculation but not necessarily the reference diet and inclusion level used. On the other hand, both AME and AMEn of CM were influenced by the reference diet, inclusion level, and assay method. Consequently, the reference diet type is an important consideration when making comparison across studies.

## Figures and Tables

**Table 1 animals-10-02132-t001:** Ingredients and composition of the reference (basal) diets.

Ingredients g/kg	Corn-SBM	Corn-CM
Corn	603.4	471.4
Soybean meal	320	0
Canola meal	0	420
Soybean oil	30	55
Dicalcium phosphate	16	19.5
Limestone	9	8.5
Titanium dioxide	5	5
Vitamin premix	5	5
Trace mineral premix	5	5
Methionine	1.5	1
Lysine	1.5	4.5
Threonine	0.5	2.0
Sodium chloride	3.1	3.1
Total	1000	1000
**Calculated nutrient content**		
Protein, g/kg	198.9	195.5
AME, MJ/kg	12.81	12.29
Ca, g/kg	8.6	8.7
Total P, g/kg	6.5	5.5
Non-phytate P, g/kg	4.2	4.2

**Table 2 animals-10-02132-t002:** Description of experimental diets and the analyzed crude protein content of the diets.

Diet No.	Test Feedstuff	Reference Diet	Test Feedstuff Added into Basal Diet, g/kg	Analyzed Crude Protein, g/kg Dry Matter
1		Corn-SBM ^1^	0	229
2	Soybean meal	Corn-SBM	300	315
3	Soybean meal	Corn-SBM	450	353
4	Canola meal	Corn-SBM	300	288
5	Canola meal	Corn-SBM	450	307
6		Corn-CM ^2^	0	232
7	Soybean meal	Corn-CM	300	306
8	Soybean meal	Corn-CM	450	354
9	Canola meal	Corn-CM	300	288
10	Canola meal	Corn-CM	450	310

^1^ Soybean meal, ^2^ Canola meal.

**Table 3 animals-10-02132-t003:** Influence of reference diet type and substitution levels on apparent metabolizable energy and nitrogen-corrected metabolizable energy values of soybean meal or canola meal determined by the difference method.

Reference Diet	Feedstuff	Substitution Level, g/kg	AME MJ/kg	AMEn MJ/kg
The main effect means for reference diet type				
Corn-SBM			11.89	10.17
Corn-CM			12.17	10.22
Pooled SEM			0.06	0.07
The main effect means for the feedstuff				
	Soybean meal		11.93	9.98
	Canola meal		12.13	10.40
Pooled SEM			0.06	0.07
The main effect means for method				
		300	12.19	10.13
		450	11.87	10.25
Pooled SEM			0.06	0.07
Simple effect means				
Corn-SBM	Soybean meal	300	11.84	9.90 ^cd^
		450	11.69	10.03 ^cd^
	Canola meal	300	12.14	10.85 ^a^
		450	11.87	9.90 ^cd^
Corn-CM	Soybean meal	300	12.31	9.63 ^d^
		450	11.87	10.37 ^abc^
	Canola meal	300	12.45	10.15 ^bcd^
		450	12.05	10.71 ^ab^
Pooled SEM			0.11	0.14
*p*-values for the main effects and interactions				
Reference diet type (RD)			<0.001	0.614
Feedstuff			0.013	<0.001
Substitution level (Level)			<0.001	0.233
RD × Feedstuff			0.606	0.922
Feedstuff × Level			0.802	0.001
RD × Level			0.176	<0.001
RD × Feedstuff × Level			0.590	0.024

*n* = 8 for the simple effects and *n* = 32 for the main effects; SEM—standard error of the means. ^a–d^ Means in a column, and within a group, but with no common superscripts are significantly different.

**Table 4 animals-10-02132-t004:** Influence of reference diet type on apparent metabolizable energy and nitrogen-corrected metabolizable energy values of soybean meal or canola meal determined by the Regression method.

Reference Diet	Feedstuff	AME MJ/kg	AMEn MJ/kg
The main effect means for reference diet type			
Corn-SBM		11.81	10.71
Corn-CM		12.06	11.07
Pooled SEM		0.03	0.03
The main effect means for the feedstuff			
	Soybean meal	12.04	10.86
	Canola meal	11.82	10.91
Pooled SEM		0.03	0.03
Simple effect means			
Corn-SBM	Soybean meal	11.94	10.70
	Canola meal	11.67	10.71
Corn-CM	Soybean meal	12.13	11.02
	Canola meal	11.98	11.12
Pooled SEM		0.04	0.04
*p*-values for the main effects and interactions			
Reference diet type (RD)		<0.001	<0.001
Feedstuff		<0.001	0.190
RD × Feedstuff		0.081	0.331

*n* = 8 for the simple effects and *n* = 16 for the main effects; SEM—standard error of the means.

**Table 5 animals-10-02132-t005:** Influence of Method on apparent metabolizable energy and nitrogen-corrected metabolizable energy values of soybean meal or canola meal.

Reference Diet	Feedstuff	Method	AME MJ/kg	AMEn MJ/kg
The main effect means for reference diet type				
Corn-SBM			11.87	10.35
Corn-CM			12.09	10.74
Pooled SEM			0.04	0.04
The main effect means for the feedstuff				
	Soybean meal		11.96	10.51
	Canola meal		12.00	10.57
Pooled SEM			0.04	0.04
The main effect means for method				
		Difference	12.03	10.19
		Regression	11.93	10.89
Pooled SEM			0.04	0.04
Simple effect means				
Corn-SBM	Soybean meal	Difference	11.77	9.96
	Canola meal		12.09	10.00
	Soybean meal	Regression	11.94	10.70
	Canola meal		11.67	10.71
Corn-CM	Soybean meal	Difference	12.01	10.37
	Canola meal		11.25	10.43
	Soybean meal	Regression	12.13	11.02
	Canola meal		11.98	11.12
Pooled SEM			0.07	0.08
*p*-values for the main effects and interactions				
Reference diet type (RD)			<0.001	<0.001
Feedstuff			0.517	0.370
Method			0.066	<0.001
RD × Feedstuff			0.797	0.656
Feedstuff × Method			<0.001	0.952
RD × Method			0.634	0.601
RD × Method × Feedstuff			0.312	0.783

*n* = 8 for the simple effects and *n* = 32 for the main effects; SEM—standard error of the means.

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
