# Peer review of "Metabolizable Energy of Soybean Meal and Canola Meal as Influenced by the Reference Diet Used and Assay Method"

_animals, 2020, doi:10.3390/ani10112132_

Round 1

Reviewer 1 Report

This article discusses the use of reference diets to determine metabolize energy of soybean meal and canola meal. The authors have used two different methods, difference and regression, to test their objective. The results indicate that the calculated metabolized energy value of feed depended on the reference diets used and a similar metabolized energy value was not observed. They claim that the regression method indicated a greater metabolize energy value for the feed compared to the different method. It is no doubt that when the composition of the reference diet is changed, the metabolize energy of the feed will change and therefore regardless of the methods used (difference or regression), the metabolized energy value will not be similar. 

The authors have definitely carried out an extensive investigation in this work but their initial hypothesis on the effects of the changes in reference diets was not corroborated strongly in this manuscript. This manuscript needs extensive revision in terms of the two methods used i.e. difference and regression. I suggest the authors elaborate on the difference and regression methods as separate sections and explain the math involved in it to calculate the metabolized energy value with statistical equations and results. A graphical representation of the statistics of the reference diet calculated with these two methods is also required. As of the time being, I don’t see this manuscript being accepted in its current form.

Author Response

Please find attached the response to the comments. Thank you.

Reviewer 2 Report

Submitted manuscript is well written and a valuable contribution. Introduction clearly presents investigated topic. Methods description is quite concise, but sufficient. Results are clearly presented and discussed. Discussion is lead well and properly referenced. Conclusions are properly drawn. Provided literature is sufficient and includes items published mostly within last ten years, however numerous older and valuable references are included.

I have only few minor remarks that should be addressed prior the publication:

Table 1 – “DCP” – please use full compound name, and “salt NaCl” –  just sodium chloride would be better. ME was not defined earlier. I am also suggesting change of calculated nutrient content line to bold.

Lines 91-92 please provide basic principle of the method.

Line 95 – please place equation in a separate paragraph.

Line 200 – CP abbreviation should be defined here

Lines 285-286 please rephrase second part of the sentence (it is hard to understand)

Author Response

Please find attached our response to your comments. Thanks.

Round 2

Reviewer 1 Report

The authors have provided relevant information which in my opinion is adequate at this time. But this manuscript could be improved if the authors include graphical representation of the statistics of the reference diet calculated.